# Thermal counting statistics in an atomic two-mode squeezed vacuum state

Maxime Perrier, Ziyad Amodjee, Pierre Dussarrat, Alexandre Dareau,
Alain Aspect, Marc Cheneau, Denis Boiron and Christoph I. Westbrook⋆

Laboratoire Charles Fabry, Institut d'Optique Graduate School, CNRS,
Université Paris-Saclay, 91127 Palaiseau cedex, France

⋆ christoph.westbrook@institutoptique.fr

## Abstract

We measure the population distribution in one of the atomic twin beams generated by four-wave mixing in an optical lattice. Although the produced two-mode squeezed vacuum state is pure, each individual mode is described as a statistical mixture. We confirm the prediction that the particle number follows an exponential distribution when only one spatio-temporal mode is selected. We also show that this distribution accounts well for the contrast of an atomic Hong–Ou–Mandel experiment. These experiments constitute an important validation of our twin beam source in view of a future test of a Bell inequality.

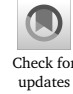
## 1   Introduction

Photon pair generation via spontaneous parametric down-conversion has been a workhorse in the field of quantum optics for many years. Numerous fundamental effects have been demonstrated ranging from squeezing to teleportation [1, 2]. Consequently, the characteristics of

down-conversion sources have been extensively studied. It has been shown theoretically that in this situation, strong photon number correlations are present between the beams, while each individual beam, after tracing over the other one, is in a thermal state [3, 4] [1].

The latter feature has been verified experimentally by measuring the pair correlation function of the individual beams, and showing that each beam exhibits the same bunching behavior as a thermal source. Some of these experiments are reviewed in Ref. [5]. Instead of correlation functions, one can alternatively examine the population distribution, or counting statistics, in a given mode to confirm its thermal nature. This type of measurement is less common in optics, although a few experiments have been carried out [6–9].

An atomic analog of the pair production process, atomic four-wave mixing, has been used in several recent experiments inspired by quantum optics [10–13]. Here too, pair correlation functions have been observed to exhibit atom bunching in individual modes [13–17]. However atomic twin-beam sources, being generated from Bose–Einstein condensates (BEC), have high spatial coherence and long coherence times, thus it is possible to directly measure populations in a single atomic mode, using either optical interactions or a micro-channel plate (MCP). These techniques have already been used in Ref. [18] to measure mode populations and in Ref. [19] to measure correlation functions up to the 6th order in a thermal atomic source. In the case of twin-beam sources, auto- and cross-correlations have been measured and shown to be consistent with the presence of a two-mode squeezed vacuum [20, 21], although in those experiments the counting statistics were not investigated. In a recent experiment, in which atomic twin beams were generated by quenching the interactions in the BEC, the mode populations were shown to be consistent with a thermal distribution [22], but the detector was not in the single mode regime.

Here, we use an MCP detector to directly measure populations in atomic twin beams. With this detector we are able to select a single atomic mode, and thus acces the full counting statistics of that mode. We show that the population is thermal, decreasing exponentially with the atom number. If the detection region is chosen to be much larger so that many modes contribute, the distribution approaches a Poissonian [7, 23]. This deformation of a thermal distribution into a Poissonian can be understood as the loss of correlation between the atoms as more modes are included. The effect has the same origin as the gradual disappearance of the bunching from a thermal source when many modes are observed [24].

The population of atomic twin beams by more than one atom in a single mode is a significant factor determining the contrast of a recent Hong–Ou–Mandel (HOM) experiment. We show that we are able to account for the observed contrast in two realizations of the HOM experiment using the observed mode populations. This type of source has also been proposed for a test of a Bell inequality using freely falling atoms. It is thus of great importance to have a good characterization of the particle source, and the experiments described below are an important part of this characterization.

## 2 Theory

A spontaneous four-wave mixing process, when pumped by two waves which are intense enough to be considered classical and undepleted, produces a set of two-mode squeezed vacuum states corresponding to a superposition of states with different numbers of pairs [4]:

$$|\psi\rangle = \sqrt{1-|\alpha|^2} \sum_n \alpha^n |n\rangle_i |n\rangle_j . \tag{1}$$

---

[1] By "thermal state", it is meant that the population distribution of the mode under consideration follows an exponential law characterized by a single parameter, which plays the role of an effective temperature.

Here, $\alpha$ is a complex number and $|n\rangle_i$ denotes a state with mode $i$ occupied by $n$ particles, and the pair $(i, j)$ corresponds to two modes which satisfy the conservation of energy and momentum, the latter condition being also known as phase matching in optics. Since many such pairs might satisfy the conservation of energy and momentum, one should strictly include many distinct mode pairs in the description of the state [17], however one can select a single spatio-temporal mode by using appropriate pinholes and filters or by post-selection. The mean number of particles in each mode is $\nu = |\alpha|^2/(1-|\alpha|^2)$. Observing the population of only one mode amounts to tracing the density matrix corresponding to Eq. (1) over the other mode. The resulting population of the successive number states is a thermal distribution given by:

$$P(n) = (1-|\alpha|^2)|\alpha|^{2n} = \nu^n/(1+\nu)^{n+1} \ . \tag{2}$$

This distribution retains its form even in the presence of a non-unit detection efficiency $\eta$. One must simply replace the mean occupation number $\nu$ by the mean detected atom number $\eta\nu$ [25, 26].

## 3 Experiment

In the experiment, we create atom pairs using four-wave mixing in a BEC subject to a moving, one-dimensional optical lattice [17]. The pairs populate a velocity distribution in the vertical direction characterized by two peaks separated by 50 mm/s, each peak having a width of about 15 mm/s; see Fig. 1. The initial condensate contains about $10^5$ atoms and about 1% of these atoms are scattered into the four-wave mixing peaks. Thus the undepleted pump approximation used above should be valid. This configuration was used as a source of pairs for two-particle interference experiments as described in Refs. [12, 27], but some fraction of the runs was devoted uniquely to observing the characteristics of the source. It is this data which is analyzed in the following.

After the nonlinear interaction, the lattice and the trapping lasers are switched off and the atoms fall onto an MCP placed 45 cm below the interaction region. In combination with a delay line anode, the detector records the positions and arrival times of single atoms [28] with an estimated detection efficiency of $\eta = 25(5)\%$. The long time of flight to the detector ensures that the positions and arrival times at the detector are proportional to the velocities of the atoms as they leave the interaction region. The data was acquired over 1876 repetitions. We chose detection volumes or "cells" in velocity space with full widths $\Delta v_x = \Delta v_y = 5.5$ mm/s and $\Delta v_z = 2.5$ mm/s for one of the two peaks in the atomic velocity distribution. The size of these cells corresponds approximately to that which optimizes the HOM interference signal which we describe in the last part of this paper. These sizes also correspond closely to the spatial widths of the measured correlation functions (see [29]). In order to increase the sample size, we examined a total of 45 contiguous cells within the peak, which filled a momentum space volume measuring $3\Delta v_x \times 3\Delta v_y \times 5\Delta v_z$ in the vicinity of the density maximum. The cells had mean detected particle numbers ($\eta\nu$) ranging from 0.08 to 0.20 per volume and per shot. To reduce the variation in mean particle number we eliminated cells with fewer than 0.135 particles per shot. This left 18 cells whose average, mean detected atom number was 0.158. We made histograms of the number of occurrences of $0, 1, 2, \ldots,$ counts in each cell, and then summed the 18 histograms to get the data shown in Fig. 2.

It is seen that the distribution is well fit by a thermal distribution. Furthermore, since the mean detected atom number is measured separately, the red line in Fig. 2 has no adjustable parameters. The figure also shows a Poisson distribution with the same mean. For detected atom numbers greater than 2, there is a clear discrepancy.

Instead of histogramming the counts in each cell separately, we can add the counts in all 18

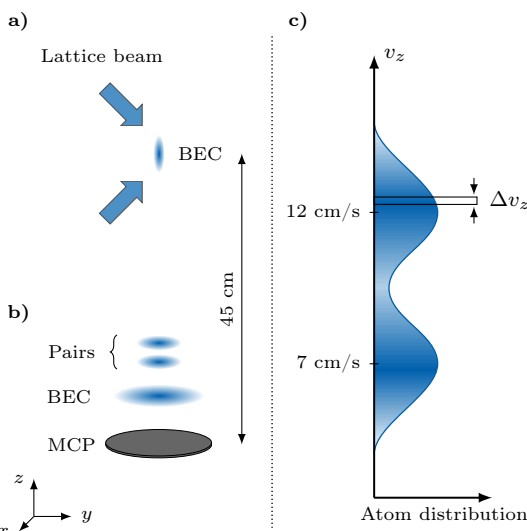

Figure 1: Diagram of the experiment. a) Atoms are emitted in pairs from a BEC confined in an optical dipole trap and subject to a moving, one-dimensional optical lattice. b) After switching off the lattice and the trap, the BEC and the clouds containing the pairs fall on the MCP detector as shown. c) We show schematically a vertical slice of the atomic velocity distribution in the clouds. The vertical axis shows the velocity relative to the BEC. We select small sections of one of these clouds and histogram the number of detected atoms over many repetitions.

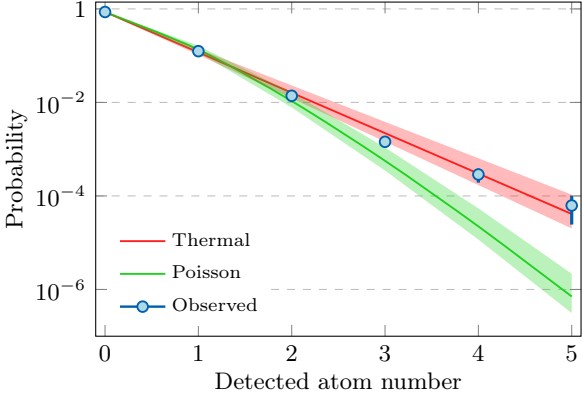

Figure 2: Counting statistics of one output mode of the atom pair source. The red line shows a thermal distribution with a mean detected atom number of 0.158, equal to that measured in the experiment. The red band shows a range of thermal distributions with detected atom numbers between 0.135 and 0.2. The green line shows a Poisson distribution with the same mean. The green band shows a range of Poisson distributions with detected atom numbers between 0.135 and 0.2. The error bars on the data points denote the statistical uncertainty (standard deviation) estimated with the bootstrapping method. Wherever the error bars are not visible, they are smaller than the point size. The populations are expressed as probabilities, therefore $P(0) < 1$.

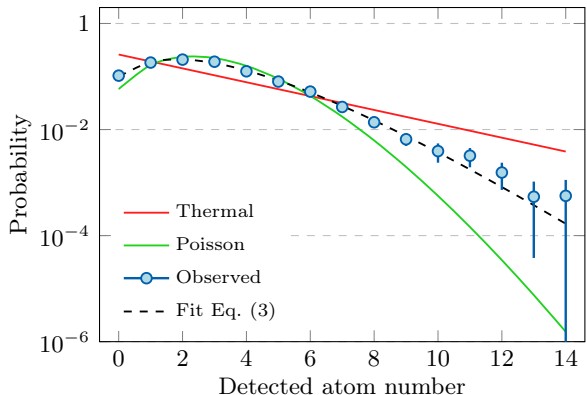

Figure 3: Counting statistics in a volume containing all 18 cells used in Fig. 2. The red line shows a thermal distribution with a mean detected atom number ($\eta\nu$) of 2.8, equal to that measured in the experiment. The green line shows a Poisson distribution with the same mean. The dashed black line is the formula in Eq. (3), with $M = 5.6$. The error bars on the data points denote the statistical uncertainty (standard deviation) estimated with the bootstrapping method.

cells together and make a histogram of those sums; see Fig. 3. This amounts to computing the average count distribution of a volume containing several modes. The distribution deviates strongly from a thermal one and shows a tendency to approach a Poissonian. It is also possible to give a formula for the expected count distribution in a volume containing a number of modes $M$ by appropriately summing $M$ identical thermal distributions. The problem is analogous to finding the counting distribution for a thermal field detected using an integration window larger than the coherence time of the field. The distribution is given by [7, 23]:

$$P_M(n) = \frac{\Gamma(n+M)}{\Gamma(n+1)\Gamma(M)}(1 + M/\nu)^{-n}(1 + \nu/M)^{-M} , \tag{3}$$

where $\Gamma$ denotes the Euler gamma function. As mentioned in the introduction, the inclusion of many modes in the counting volume leads to the counting events becoming uncorrelated, behaving as a simple rate process and following a Poisson distribution in the limit of large $M$.

In the above formula, the parameter $M$ is often referred to as the degeneracy parameter [23] and need not be an integer. Since this distribution corresponds to the sum of thermal distributions, it remains valid even in the presence of a non-unit detection efficiency, provided that $\nu$ be replaced by $\eta\nu$. Figure 3 shows a fit to this formula using $M$ as the only adjustable parameter. The data is well fit by Eq. (3) and $M = 5.6(7)$. That the value of $M$ is smaller than the number of volumes over which we averaged reflects the fact that individual modes have a smooth shape and in order to observe single-mode statistics as in Fig. 2, the detection volume must be smaller than the characteristic mode size to avoid including contributions from neighboring modes.

## 4 HOM experiment

Having established the populations of the individual modes, we now turn to analyzing data from an HOM experiment, in which we recombine the two (correlated) modes on a beam splitter and look for two-particle interference. The data was acquired at the same time as that used in the experiment of Ref. [27]. One atom from a pair is sent to each input port of the beam splitter. The HOM effect is the dramatic decrease in the cross correlation between the

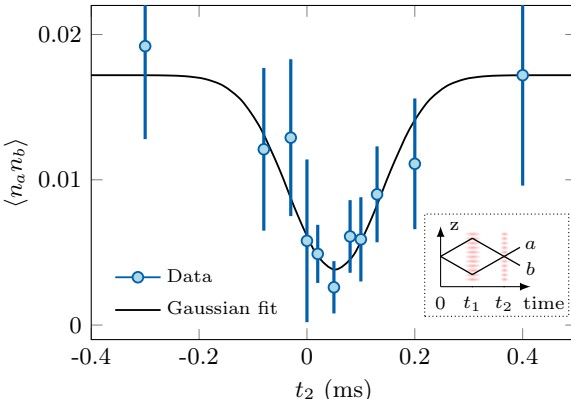

Figure 4: The cross correlation between the two output modes $a$ and $b$ as a function of the time $t_2$ of the beam splitter (the position of $t_2 = 0$ is arbitrary). The line shows a fit to a Gaussian function. The fitted visibility is 0.78(6) and the RMS width is 86(15) µs. The error bars on the data points denote the statistical uncertainty (standard deviation) estimated with the bootstrapping method. The inset shows our HOM interferometer in the center-of-mass frame of the atom pair. Laser standing waves act as deflectors and beam splitters and the outputs $a$ and $b$ correspond to small regions at the detector.

output ports when the two spatio-temporal modes are overlapped. We characterize the effect by the visibility $V$, defined as the maximum fractional decrease in the correlation. In Ref. [12] we showed that the observed visibility was limited by the presence of multiple atoms in the interferometer. More specifically, we showed that it was consistent with that predicted by the correlations in the two beams before the beam splitter. This analysis made few assumptions about the nature of the source. In the following we will examine a different analysis which assumes that the source produces a two-mode squeezed vacuum state, and thus allows us to relate the HOM visibility to the mean mode population as approximately measured in Fig. 2. In Ref. [30] it is shown that in the case of a two-mode squeezed vacuum state, the visibility is given by:

$$V = 1 - \left( 2 + \frac{1}{2\nu} \right)^{-1} . \tag{4}$$

The visibility approaches unity in the limit of small average particle numbers and it approaches ½ for large particle numbers. In contrast, two uncorrelated thermal states would lead to a maximum visibility of ⅓. Observing the HOM visibility gives additional information compared to the single mode counting statistics above, because it is sensitive to the correlation between the two modes. We emphasize that the HOM visibility is independent of the detection efficiency; the quantity $\nu$ in Eq. (4) is the actual number of atom pairs produced in a single mode pair by the source.

The experimental procedure was described in Ref. [27]. A simple diagram is shown in the inset to Fig. 4. Briefly, after a time of free flight $t_1 \simeq 1$ ms, the two atoms of a pair are Bragg diffracted on a laser standing wave to bring them back together. At a time designated $t_2$, about 2 ms after the pair was produced, another standing wave acts as a beam splitter mixing momentum states. We observe atoms at each output, labeled $a$ and $b$, and compute the correlation $\langle n_a n_b \rangle$, where the angle brackets denote an average over 700 to 1 000 experimental realizations per set time $t_2$. We use the same integration volume as in Ref. [27] to count the atoms: a cylinder oriented along the $z$ axis with a height 2.6 mm/s and a diameter 4.1 mm/s. To observe the HOM effect, we scan the application time $t_2$ of the beam splitter to vary the overlap between the two spatio-temporal modes. The data showing the correlation between

Table 1: Comparison of the visibilities in two realizations of the atomic HOM experiment. The column labeled $\nu$ is an estimate of the number of atoms per mode taking into account the 25 % quantum efficiency of the detector. The uncertainty in $\nu$ is that due to the detection efficiency (see text). The column $V_{\text{pred}}$ is the visibility predicted by Eq. (4), assuming that the number of atoms per mode is $\nu$. The visibility observed in the experiment is $V_{\text{obs}}$, and its uncertainty results from a fit such as shown in Fig. 4.

| experiment | $\nu$ | $V_{\text{pred}}$ | $V_{\text{obs}}$ |
|---|---|---|---|
| Ref. [12] | 0.8(2) | 0.62(2) | 0.65(7) |
| present work | 0.33(7) | 0.72(2) | 0.78(6) |

the two output ports is shown in Fig. 4. The figure shows the expected dip with a visibility of 0.78(6).

To make contact with the prediction of Eq. (4), we need to estimate $\nu$, the true population of the interfering modes, a quantity that we cannot measure directly in the experiment. A simple estimate consists in using the average count rate in one of the modes and correcting by the detection efficiency. We show a comparison between the observed and predicted visibilities using this estimate. The uncertainty in the predicted visibility is determined by the 20% uncertainty in the detection efficiency. Any error due to the detection efficiency is the same for the two realizations. This uncertainty is an underestimate because we cannot be sure that the number atoms in a given cell is the true number of atoms in the mode. As for the detection efficiency, this error would be the same in the two realizations. Other sources of uncertainty: statistical errors in the number of counts, beam splitter imperfections etc. contribute significantly less than the quantum efficiency to the uncertainty in the predicted visibility.

## 5 Conclusion

The agreement seen in Table 1 together with the population measurements support the conclusion that, after mode selection at the detector, our source does produce a two-mode squeezed vacuum state of Eq. (1). The experiment of Ref. [27] observed interference between different sets of modes. To extend that experiment to a test of a Bell inequality, one condition to be met is that the populations of the modes be low enough to prevent contamination from additional particles. Knowing that our state has the form of Eq. (1) allows us to calculate the threshold and optimize the particle number in advance [31].

## Acknowledgements

**Funding information**   We acknowledge funding by the EU through the Marie-Curie Career Integration Grant 618760 (CORENT), the QuantERA grant 18-QUAN-0012-01 (CEBBEC) and the ANR grant 15-CE30-0017 (HARALAB).

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
