# Peer review of "Thermal counting statistics in an atomic two-mode squeezed vacuum state"

_SciPost Physics, doi:SciPost Phys. 7, 002 (2019)_

## Round 1 · Referee Report · Anonymous (Referee 1) · 2019-5-3

Strengths

1 - Very clear and well written manuscript with appropriate figures.
2 - Basic theory is outlined sufficiently.

Weaknesses

1 - Novelty of results is not made clear.

Report

Perrier et. al. present an investigation of four-wave mixing in an optical lattice, specifically focused on characterizing the dynamically produced state as a two-mode squeezed vacuum. The main result is the measurement of the single-mode distribution function P(n) of the mode population n, which they demonstrate is consistent with the thermal distribution predicted by the entangled two-mode squeezed state. They also present data from an atomic Hong-Ou-Mandel experiment (which they previously demonstrated elsewhere) and show that the visibility of the HOM `dip' is consistent with a simple model which assumes a two-mode squeezed vacuum as the input state and is thus characterized solely by the mean mode occupation <n>.

I have a few comments regarding the material in the paper. Overall, I find the paper is not clear in establishing what they regard as novel here. Specifically, whilst atomic four-wave mixing experiments are ideal to study P(n) (due to, e.g., the typically larger mode occupation compared to quantum optics experiments), this result has already been published for a two-mode squeezed vacuum in another atom-optics setup in arXiv:1807.07504 (as they note in the manuscript). Perhaps the authors might comment on whether there is a particular (in principle) advantage to the platform presented here: For example, does the tunability of the mode population in the optical lattice setup mean it might be easier to study the distribution P(n) beyond the undepleted pump regime presented here?

Further to this, I wonder if the authors could comment on whether there is the possibility of extracting further data to support their conclusion that the source state is indeed a two-mode squeezed vacuum. In particular, given the ability to extract histograms P(n) for a single mode (with additional averaging), is it possible to investigate the two-mode distribution P(n_1 + n_2) and observe evidence of an odd-even oscillatory behavior?

Overall, the paper is well written and the included results are discussed sufficiently. However, I would like the authors to respond to these points before I make a recommendation for publication.

Requested changes

1 - The authors should cite Phys. Rev. Lett. 118, 240402 (2017) as well as Ref. 19. Here, the same group reports characterization of 3rd and 4th order correlations which are consistent with a two-mode squeezed vacuum source.

2 - The authors should give an estimate of the total depletion of the condensate. I understand that the experimental details can be found in previous publications, but the case for the resulting state being two-mode squeezed vacuum is based on the undepleted pump assumption!

3 - The observed and predicted visibility for this work are only just in agreement due to the error bars (attributed to fitting) in Table 1. Perhaps the authors could give a rough estimate of what additional sources of error (e.g., technical noise due to imperfect beam-splitters etc) might contribute so the result can be viewed with more confidence.

  • validity: high
  • significance: low
  • originality: low
  • clarity: high
  • formatting: excellent
  • grammar: perfect

Author:  Christoph Westbrook  on 2019-06-17  [id 542]

(in reply to Report 1 on 2019-05-03)
Category:
answer to question

The reviewer is correct that the tunability of our source might allow us to examine deviations from the undepleted pump approximation. However, the most important point for us in this work is to characterize as well as possible the source we intend to use in a Bell test. A novelty compared to arXiv 1807.07504 is that we are able to isolate a single mode and measure the full counting statistics. We have added two sentences to the introduction to emphasize these points.

The reviewer asked about whether is it possible to investigate the two-mode distribution P(n_1 + n_2) and observe evidence of an odd-even oscillatory behavior. We examined this distribution, but unfortunately, the available data was too noisy to draw any clear conclusion.

The other requested changes have been implemented.

---

## Round 1 · Referee Report · Anonymous (Referee 2) · 2019-5-28

Strengths

1- well written. 2-interesting topic.

Weaknesses

1-Motivation not clearly stated.
2-see report.

Report

This paper reports on the counting statistics of a source of atoms produced via a FWM process in an optical lattice. The authors aim is to show that the properties of their source match that of a set of two-mode squeezed vacuum states. They do this in two ways - confirming that the counting statistics of one of the correlated modes, follows thermal statistics, provided the binning used is less than the mode size, and secondly by matching the observed visibility in a HOM dip to that predicted for a two-mode squeezed vacuum source. Ultimately, the authors are interested in using their system to test Bell’s inequality. In order to do this difficult experiment efficiently the mode occupation is a critical parameter, as any such violation will depend on this. This is what motivates their experiment, although they leave it to the paper’s conclusion to elucidate this.

I believe the paper should be published, but I have a number of questions that should be addressed.

Requested changes

• Firstly, the paper needs better motivation. Please add a para about how the results here will specifically aid in the future Bell test.
• Page 3, Can the authors give an estimate of the expected mode size? Otherwise, how do we know that Figure 2 shows what the authors expect?
• Page 3, The authors say they eliminate cells with less than 0.135 particles per shot. This seems totally arbitrary, especially since the remaining cells only have 0.158 particles per shot (i.e. about 15% more). Can the authors please justify their reasoning for removing this data. Do their results with the additional cells, follow the same trend, but with larger error bars?
• What is meant in Figure 2: “Note that P (0) < 1, since the populations have been normalized.”
• Page 4 the authors state “The good agreement indicates that any distortion of the distribution due to detector saturation is negligible at the maximum present flux level (5 particles in 0.25 ms over an area of 3 mm2).” This seems an odd statement to me, basically they are saying that because they observe what is predicted then “distortions” are thus not present. What if the distortions cause them to observe something that agrees with what they expect? The authors should just state that at this flux rate these type of detectors are expected to operate linearly (without saturation) – with a reference if possible.
• Page 4, I believe equation 3 is incorrect, should the (1+n/v)^v be (1+M/v)^-n ?
• Page 4, the authors state “It is also possible to give a formula for the expected count distribution in a volume containing a number of modes M by appropriately summing M identical thermal distributions.” – So M is both the number of modes and the number of thermal distributions – I am confused.
• Page 4, Can the authors not make an estimate about how many modes they average over based on what they know about their source. In such case is M=5.6 reasonable in that context?
• Page 6, what do the authors mean by “the average singles count rate”

Page 1, para 3 line 3 Replace “But” with “However”
Page 3, para 2 line 14 replace “dispersion” with “variation”
Figure 2, and anywhere else Replace “Count distribution” with “Counting statistics”
Page 5, Second last line “Two uncorrelated thermal…” -> “In contrast, two uncorrelated…”
Fig 4 replace “contrast” with “visibility”

  • validity: high
  • significance: good
  • originality: good
  • clarity: high
  • formatting: excellent
  • grammar: good

Author:  Christoph Westbrook  on 2019-06-18  [id 543]

(in reply to Report 2 on 2019-05-28)
Category:
answer to question

The reviewer has made several good points and we have modified the manuscript accordingly. Below we treat those points which require more than a simple modification of the text.

The reviewer asked about other estimates of the expected mode size. There are two ways to estimate the mode size which do not depend on observing the counting statistics. The first, as we mention in the text, is to use the volume which optimizes the HOM signal. Another is to observe the auto-correlation function of a beam, exactly as was done in Refs. [17] and [21] (we use the reference numbers in the revised manuscript). These measurements can be found in the thesis of P. Dussarrat. Dussarrat found delta x=delta y ~ 2.7 mm/s and delta z ~ 6.9 mm/s, that is, within 20% of the values used to bin the data in Fig. 2. The Dussarrat reference has been included in the text.

There was a remark about the apparently arbitrary elimination of cells with less than 0.135 particles per shot. Ideally, we want to average cells with the same average count rate, so that they all contribute with the same exponential function. So severe a selection would drastically reduce our statistical power. We therefore must make a compromise between severe selection and small uncertainties. In the accompanying pdf file we show examples of the counting statistics using different numbers of cells plotted in the same way as in Figs. 2 and 3. One can see that the general trend is independent of the degree of selection. Our choice was made to make sure the thermal and Poissonian predictions were well separated, but other choices are possible.
The reviewer was bothered by our statement that detector distortions are not present. There is very little quantitative published information about saturation in such detectors. Our own experience is that the noticeable saturation occurs at substantially higher count rates.
Even though our analysis assumes the detector is linear, we wanted to use the fact that the counting statistics conform to our expectations to support our increased confidence in the detector. Yes, strictly speaking the logic is circular. The point is not essential and we have deleted it in the new version.

The reviewer asked about the meaning of the parameter M and whether we could estimate its value. If we interpret M as a number of modes, and if each mode has a thermal distribution with the same mean, then the expected distribution is the sum of M thermal distributions. Of course we do not have an exact integral number of modes. What we can be sure about is that 18 > M > 1 (for the case of 18 cells). The distributions are smooth and do not correspond strictly to modes in the sense of orthogonal spatial functions. Thus more careful discussions refer to M simply as a "degeneracy parameter".

The expression "singles count rate" is a bit of jargon used to distinguish the rate in a single detector from the "coincidence count rate", or correlation. The expression has been removed.

Attachment:

selectionboites_copy.pdf

---

## Round 2 · Referee Report · Anonymous (Referee 1) · 2019-6-18

Report

I thank the authors for their response, in which they have satisfactorily addressed my concerns and questions.

---

## Round 2 · Referee Report · Anonymous (Referee 2) · 2019-6-19

Report

The authors have addressed my main criticisms, and I thank them for the additional data plots showing that the trend they observe is present independent of bin selection.

---

## Round 2 · Author Response

Dear Editors
We thank the reviewers for their careful reading of the manuscript. In cases where we felt it was needed, we responded to the reviewers questions. Otherwise we have implemented the suggestions as listed below.

---

## Round 2 · List of Changes

1. We have added two sentences to the introduction to emphasize the novelty of this work: "With this detector we are able to select a single atomic mode, and thus access the full counting statistics of that mode. We show that the population is thermal, decreasing exponentially with the atom number."

  2. We have added the reference suggested by the reviewer, as well as another one on the same subject (Kheruntsyan et al. 2012). "In the case of twin-beam sources, auto- and cross-correlations have been measured and shown to be consistent with the presence of a two-mode squeezed vacuum [20,21], although in those experiments the counting statistics were not investigated."

  3. Two sentences have been added to "Experiment" concerning the depletion: "The initial condensate contains about $10^5$ atoms and about 1% of these atoms are scattered into the four-wave mixing peaks. Thus the undepleted pump approximation used above should be valid."

  4. We have amended table I to include the primary, known uncertainty, the quantum efficiency of the detector. We have also modified the discussion of this uncertainty to give a better indication of the sources. "We show a comparison between the observed and predicted visibilities using this estimate. The uncertainty in the predicted visibility is determined by the 20\% uncertainty in the detection efficiency.Any error due to the detection efficiency is the same for the two realizations. This uncertainty is an underestimate because we cannot be sure that the number atoms in a given cell is the true number of atoms in the mode. As for the detection efficiency, this error would be the same in the two realizations. Other sources of uncertainty, statistical errors in the number of counts, beam splitter imperfections etc. contribute significantly less than the quantum efficiency to the uncertainty in the predicted visibility."

  5. We have added a sentence in the abstract and two sentences in the introduction concerning the importance of characterizing the source for a Bell test. Abstract: " These tests constitute an important validation of our twin beam source in view of a future test of a Bell inequalities." Last paragraph of the introduction: "This type of source has also been proposed for a test of a Bell inequality using freely falling atoms. It is thus of great importance to have a good characterization of the particle source, and the experiments described below are an important part of this characterization."

  6. We have added a reference to the thesis of P. Dussarrat in which an estimate of the mode size using the autocorrelation is given. (see response to reviewer 2)

  7. We have amended the caption to Fig. 2 to indicate that the populations are expressed as probabilities.

  8. We have deleted a sentence concerning confirmation of the linearity of the detector (see response to reviewer 2).

  9. The phrase "singles count rate" has been changed simply to "count rate" (see response to reviewer 2).

  10. Page 1, para 3 line 3 Replace “But” with “However”: done.

  11. Page 3, para 2 line 14 replace “dispersion” with “variation”: done.

  12. Figure 2, and anywhere else Replace “Count distribution” with “Counting statistics”: done.

  13. Page 5, Second last line “Two uncorrelated thermal…” -> “In contrast, two uncorrelated…”: done.

  14. Fig 4 replace “contrast” with “visibility”: done.

---

## Editorial Decision

published